# Diagnosis of colorectal cancer using residual transformer with mixed attention and explainable AI

**Poonam Sharma[1], Bhisham Sharma ![ORCID][2], Ajit Noonia ![ORCID][3], Dhirendra Prasad Yadav[4], Panos Liatsis ![ORCID][5]***

**1** Chitkara University Institute of Engineering and Technology, Chitkara University, Rajpura, Punjab, India, **2** Centre of Research Impact and Outcome, Chitkara University, Rajpura, Punjab, India, **3** Department of Computer Science and Engineering, Manipal University Jaipur, Jaipur, Rajasthan, India, **4** Department of Computer Engineering & Applications, G.L.A. University, Mathura, Uttar Pradesh, India, **5** Center for Cyber Physical Systems, Department of Computer Science, Khalifa University, Abu Dhabi, United Arab Emirates

\* Panos.liatsis@ku.ac.ae

## Abstract

Colorectal cancer (CRC) is the leading cause of cancer disease and poses a significant threat to global health. Although deep learning models have been utilized to accurately diagnose CRC, they still face challenges in capturing the global correlations of spatial features, especially in complex textures and morphologically similar features. To overcome these challenges, we propose a hybrid model using a residual network and transformer encoder with mixed attention. The Residual Next Transformer Network (RNTNet) extracts spatial features from CRC images using ResNeXt. ResNeXt utilizes group convolution and skip connections to capture fine-grained features. Furthermore, a vision transformer (ViT) encoder containing a mixed attention block is designed using multiscale feature aggregation to provide global attention to the spatial features. In addition, a Grad-CAM module is added to visualize the model's decision process to support oncologists with a second opinion. Two publicly available datasets, Kather and KvasirV1, were utilized for model training and testing. The model achieved classification accuracies of 97.96% and 98.20% on the KvasirV1 and Kather datasets, respectively. Model efficacy is also further confirmed by ROC curve analysis, where AUC values of 0.9895 and 0.9937 on the KvasirV1 and Kather datasets are obtained, respectively. Comparative study findings support that RNTNet delivers improvements in accuracy and efficiency compared to state-of-the-art methods.

## 1. Introduction

Colorectal cancer is a form of cancer that originates as a polyp, which, due to genetic changes, may develop into cancer. Various factors, such as age, heredity, inflammatory bowel disease, and genetic mutations, could be leading causes of this disease

**Data availability statement:** The datasets used in the study can be accessed using URL: https://zenodo.org/records/53169, https://data-sets.simula.no/kvasir/.

**Funding:** The author(s) received no specific funding for this work.

**Competing interests:** The authors have declared that no competing interests exist.

[1]. Some habits, such as smoking, consuming processed meat, and excessive alcohol consumption, also increase the risk of developing CRC [2]. At the early stages, the disease is not noticeable. However, patients may experience tiredness, changes in bowel habits, blood in stool, stomach pain, or weight loss. Some non-invasive clinically practised techniques are the Fecal Occult Blood Test (FOBT) and the Fecal Immunochemical Test (FIT), which use stool samples to detect the occurrence of this disease [3]. An invasive option is colonoscopy, where a camera is inserted into the colon to detect the presence of polyps or tumours. Sigmoidoscopy is another invasive method of colon examination [4]. Eating fibre-rich food, an active lifestyle, quitting smoking, and having regular checkups can lower the risk and prevent CRC [5].

FIT and FOBT are common ways to detect the disease; however, these methods may not be able to detect early signs of colorectal cancer, while blood may not be present in stool, which could lead to misdiagnosis [6]. Similarly, colonoscopy and sigmoidoscopy can be uncomfortable and intimidating for patients, potentially delaying disease diagnosis [7]. Recently, advanced methods utilising biological markers, improved imaging (e.g., radiomics), and artificial intelligence (AI) have been used to detect CRC [8]. AI methods such as machine learning (ML) and deep learning (DL) are being used by doctors to support early cancer detection and subsequent treatment planning. These technologies enhance the speed and accuracy of processing medical images and patient data, as well as performing genetic data analysis [9]. For example, polyps in colonoscopy images can be detected and classified based on DL models, e.g., Convolutional Neural Networks (CNN). Microscopically, other models could be used to assist in interpreting tissue samples and analyzing CT scans. Tissue sample classification into classes, e.g., non-cancerous, pre-cancerous, and cancerous, can assist doctors and pathologists in achieving more accurate conclusions [10].

Previous efforts [11] attempted to tackle this challenge using a vision transformer model, which incorporates multi-head self-attention (MHSA) to provide for global correlations of the spatial features. However, the computational costs of the MHSA are quadratic in time, and require a high volume of training data for appropriate model generalization. Moreover, the classical transformer model lacks local spatial features; consequently, local and complex textural features are not adequately captured. Further, their model was tested on the binary classification task. In this work, the DL model of RNTNet is introduced in CRC diagnosis. This contains the ResNeXt architecture for extraction of high dimensional local spatial features. Further, a mixed attention-based transformer encoder is applied to the flattened features obtained from the ResNeXt block, which provides local and global attention to the feature map through the Overlapping Block Attention (OBA) and Grid Attention (GBA) blocks, respectively. Due to this, the model is capable of effectively focusing on the edge and boundary regions of the colon cancer regions. In addition, computational costs of the mixed attention block are $O(M^2D^2)$, where M is the feature map size and D is the dimension, while those of MHSA in the classical ViT are $O(M^4D)$, which is much higher than the proposed model. Further, we tested RNTNet on the diverse datasets KvasirV1 and Kather, which contain 8 types of diseases, i.e., multi-class classification.

The contributions of this research are as follows:

(1) We utilized the ResNeXt, a group convolution-based residual network for high-dimensional spatial feature extraction and the ViT encoder to provide global contextual information for precise CRC diagnosis.

(2) Further, in the ViT encoder, we integrated overlapping block attention and grid attention to effectively capture local and global spatial and contextual information of the colorectal cancer regions.

(3) A Grad-Cam module was incorporated in the proposed system to provide explainability on the decision process of the model to support its use as a second opinion to oncologists.

(4) We used two datasets, KvasirV1 and Kather, to evaluate the performance of the model and compared it with state-of-the-art (SOTA) methods.

The remainder of the manuscript is arranged in the following way.

Section 2 provides a brief overview of colorectal cancer diagnosis using deep learning approaches. The RNTNet architecture is introduced, and its details are explained in Section 3. Section 4 provides quantitative results on model performance. A detailed discussion, which includes comparison with SOTA approaches, training stability analysis, explainability, and complexity, is provided in Section 5. The conclusions of the research, along with its drawbacks and future directions, are presented in Section 6.

## 2. Literature review

Kiehl et al. [12] used CNN in histological whole slide images (WSI) of 2,431 primary tumours. The area under the receiver operating characteristic (AUROC) curve was 67.0% on the clinical database of 582 images and 71.0% on the test dataset. Wang et al. [13] proposed the Efficient Multi-Task Synergetic Network (EMTS-Net), which was used in simultaneous segmentation and classification of colonoscopy images. The proposed model achieved an average AUC of 0.913 and an average accuracy of 0.924 in polyp classification. Zamanitajeddin et al. [14] proposed a novel technique to improve biological pathways and critical mutation prediction for CRC using DL. The method demonstrated an AUROC of 99% and an area under the precision-recall curve (AUPRC) of 98% on an external dataset. Sharkas and Attallah et al. [15] proposed a DL model, Color-CADx, in CRC. The model was trained on the NCT-CRC-HE-100K and Kather_texture_2016 datasets and achieved accuracies of 99.3% and 96.8%, respectively.

Talebi et al. [16] proposed an ML model, which was trained on a dataset obtained from 1,127 patients, treated at Taleghani Hospital, to forecast metastases in patients with CRC. Five-fold cross-validation was used to assess model performance. This was reported to achieve an AUC of 100% and an accuracy of 99.2% using neural networks and random forests. Li et al. [17] introduced a few-shot learning technique in classifying histological images of colorectal cancer into benign and malignant groups. The model achieved an accuracy of 92%. Narasimha et al. [18] enhanced the diagnosis of colorectal cancer using a CNN. The model was shown to perform better in combination with ResNet and DenseNet and achieved an accuracy of 97.93%. Prezja et al. [19] proposed a hybrid approach to enhance the processing of histological images of colorectal cancer through EfficientNetV2. The model achieved an accuracy of 96.74% on an external test set and 99.89% on the internal test set. Ayana et al. [20] designed two main modules. The first was ViTCol, an improved ViT for the classification of endoscopic pathology findings. The second module was PUTS, a model intended to enhance the boundary delineation of polyps in endoscopic images through precise polyp segmentation. ViTCol attained an AUC of 0.9999 on the Kvasir dataset. PUTS demonstrated segmentation with mean intersection over union (mIoU) values of 0.8673 on the Kvasir-SEG and 0.9092 on the clinical datasets.

Lo et al. [21] utilized colonoscopy images and developed a ViT-based model for detecting microsatellite instability (MSI) in colorectal cancer. The ViT model achieved an accuracy of 84% and an AUC value of 0.86. The model used 2,017 colonoscopy images (427 MSI-H, 1,590 MSS) in the study. Chughtai et al. [22] introduced DeepCon, a model for the

classification of CRC histopathological images. The model was trained effectively on a dataset of 5,000 CRC images from eight classes. The model proved its efficacy in CRC diagnosis with an accuracy of 98.4%.

Tang et al. [23] developed a multimodal guided complementary network (MGCNet), which employs multi-sequence MRI for patients undergoing hepatic resection for colorectal cancer liver metastasis (CRLM). The model showed an accuracy of 0.79 and an AUC of 0.84. Lv et al. [24] proposed a disentangled representation-based multimodal fusion (DRMF) framework for Kirsten rat sarcoma (KRAS) mutation. During testing on a dataset of 111 patients, the model revealed an accuracy of 0.876 and an AUC of 0.865. Fu et al. [25] suggested an interpretable network (IPNet) for full-stage colorectal disease diagnosis. The model was trained on two large datasets of 11,072 endoscopic ultrasound images for rectal tumours, attaining an accuracy of 89.62%, and 129,893 endoscopic optical images for colorectal lesion classification, attaining an accuracy of 93.15%. Cai et al. [26] developed CHNet and proposed a multi-task global-local collaborative hybrid network based on T2-weighted MRI images of 320 patients to evaluate the role of KRAS mutations in colorectal cancer. CHNet achieved an accuracy of 88.93%. Kumar et al. [27] developed the CMNV2 model, combining the modified MobileNetV2 (MMNV2) with a CNN for fast feature extraction (CAFFE) framework using a dataset consisting of 10,000 images. The model achieved an accuracy of 99.95%.

## 3. Methodology

In this research, we propose RNTNet for the diagnosis of colorectal cancer. An overview of the system is shown in Fig 1. We combine overlapping block attention (OBA) and grid attention (GA) to provide local and global attention to the spatial feature map. In traditional ViTs, multi-head self-attention (MHSA) is applied, which ignores local feature dependencies and has a quadratic computational cost. The necessity and efficacy of this design are demonstrated by the ablation study (Table 6), which shows that OBA and GA together significantly outperform MHSA. Although there are ResNeXt + ViT hybrids, they usually employ standard MHSA, which is computationally costly $O(M^4D)$. Our suggested mixed attention block provides an accurate yet computationally efficient solution by reducing complexity to $O(M^2D^2)$.

The input images are fed to the ResNeXt for feature extraction. Next, the extracted features are flattened, and tokens are generated for the ViT encoder. In the ViT encoder, we utilized mixed attention mechanisms for global correlation of the

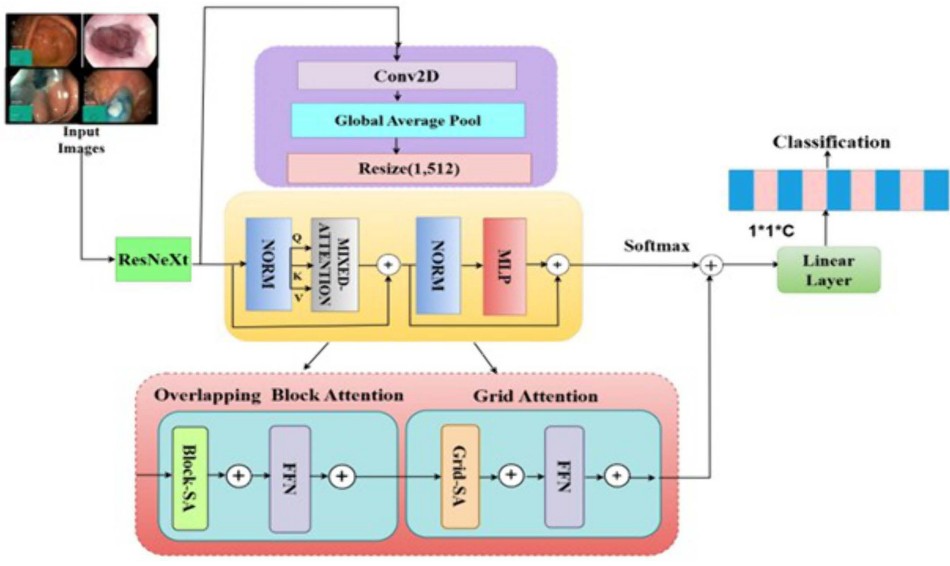

**Fig 1. The RNTNet architecture for the diagnosis of colorectal cancer.**

spatial features. The output of the attention modules is fed into feed-forward networks (FFN) and subsequently through the classification head, where a Softmax layer classifies colorectal cancer tissues.

### 3.1. The ResNeXt block

ResNeXt is a residual network known for its ability to extract high-dimensional features. It has 50 layers and skips connections to help avoid the vanishing gradient problem, as shown in Fig 2. The ResNeXt block processes the input tensor through group convolutions. Let the input be $X \in R^{H \times W \times C}$, where $H$, $W$, and $C$ represent the height, width, and number of channels, respectively. Channel reduction is performed through a 1x1 convolution that results in $X_1 = Conv\,1 \times 1(X)$, spatial feature extraction through a 3x3 group convolution as $X_2 = Conv\,3 \times 3(X_1)$, and channel expansion through another 1x1 convolution that results in a feature map $X_3 = Conv\,1 \times 1(X_2)$. This convolution result is passed into a skip connection as in Equation 1, where $w$ and $b$ are respectively the weight and bias; downsampling occurs when the stride is greater than 1. This ensures that the block maintains its residual connection, thereby helping to mitigate the vanishing gradient problem.

$$Skp = Conv(w * X + b) \tag{1}$$

The final output of the ResNeXt block is passed through a ReLU activation function to introduce non-linearity. Further, multiple ResNeXt residual blocks are aggregated in four stages with different filter sizes in the ResNeXt50 model. Stage 1 consists of 128 filters and three blocks, Stage 2 has 256 filters and four blocks, Stage 3 has 512 filters and six blocks, and Stage 4 has 1024 filters and three blocks. The output feature map $X'$ at the end of this network can be described as in Equation 2.

$$X' = ReLU(X_3 + Skp) \tag{2}$$

A flattening layer is then applied to reshape this multi-dimensional tensor into a vector, as in Equation 3.

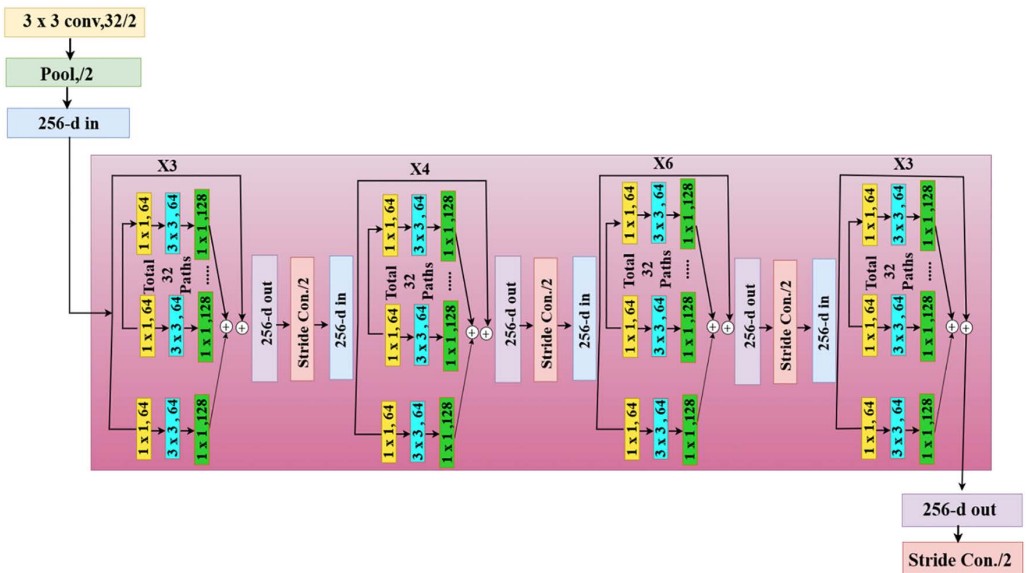

**Fig 2. The architecture of the ResNeXt block.**

$$Y = \text{Flatten}(X')$$ 
(3)

To transform the flattened output Y into patches, we apply Equation 4 to reshape it into a multi-dimensional tensor, e.g., 2D spatial patches.

$$X = \text{Reshape}\,(Y, (N, P_H, P_W, C))$$
(4)

where the height of each patch is $P_H$, the width of each patch is $P_W$, C is the channels, and N is the number of patches.

### 3.2. Vision transformer block

The ViT is responsible for providing long-range dependencies to the spatial features. In the ViT, attention scores are calculated using the query (Q), key (K), and value (V), as shown in Equation 5.

$$\text{Atten}(Q, K, V) = \text{softmax}\left(\frac{QK^T}{\sqrt{d_k}}\right)$$
(5)

where $d_k$ is the dimensionality of the key vectors. The output $X_{out}$ is calculated via layer normalization of the sum of the input X after passing through the FFN, the attention output and a residual connection to the input itself. This mechanism captures long-range dependencies and enhances model performance in colorectal cancer diagnosis. The $X_{out}$ is mathematically calculated as in Equation 6.

$$X_{out} = \text{Norm}\,(X + \text{Atten}(Q, K, V) + \text{FFN}(X))$$
(6)

where *Norm* is layer normalization applied to stabilize the learning process.

### 3.3. Overlapping block attention

The computational cost of MHSA in the classical ViT is of quadratic time. In addition, this focuses solely on global contextual information. In this study, an overlapping block attention mechanism was designed to enhance the analysis of colorectal cancer regions. This incorporates convolutional layers for attention computation, including a 3x3 convolution, followed by ReLU activation for processing local spatial areas of the input. The spatial dimensions of the resulting feature maps are transformed into a single vector via global average pooling (GAP). Subsequently, an FFN and a fully connected layer are applied to refine the pooled spatial features. The input is divided into blocks of size $B \times B$ as shown in Equation 7, where $B$ is the size of the block, and $s$ is the stride.

$$n = \left(\frac{H - B}{s} + 1\right) \times \left(\frac{W - B}{s} + 1\right)$$
(7)

For overlapping block $X_b$, we calculate the Q, K and V for the self-attention mechanism, as shown in Equation 8.

$$Q = X_b W_Q, \ K = X_b W_K, \ V = X_b W_V$$
(8)

where $W_Q, W_K, W_V \in R^{C \times D}$ are learnable matrices. Next, the block's attention is computed using Q, K, and V, as shown in Equation 9.

$$A = \left(\frac{QK^T}{\sqrt{D}}\right) V$$
(9)

$A \in R^{B^2 \times B^2}$ is the attention weight, and $D$ is the feature dimension after the reduction. The sum of weighted values is calculated as in Equation 10. The output of each block is $Z_b \in R^{B^2 \times D}$.

$$Z_b = AV \tag{10}$$

Finally, we aggregate the attention from overlapping blocks, as shown in Equation 11. For overlapping blocks, a weighted average is applied to integrate them and a function $W_b$ applied to emphasize prominent features.

$$Z(i,j) = \frac{\Sigma_b W_b \ (i,j) Z_b \ (i,j)}{\Sigma_b W_b \ (i,j)} \tag{11}$$

### 3.4. Grid attention

Grid attention calculates the attention within a grid, which allows the transformer to capture broad spatial relationships between pixels. In addition, it focuses on local neighbours and aggregates information across the grid, which makes it effective for the extraction of fine-grained spatial features and identification of long-range dependencies. GA includes an additional downsampling step in the convolution, which downsamples the input feature map using a stride of 2. Furthermore, it captures long-range dependencies by dividing the image into grids and applying a self-attention mechanism as shown in Equation 12.

$$X_g \in R^{P^2 \times C} \tag{12}$$

where $P$ is the patch size of a non-overlapping grid having a size of $G \times G$, and the self-attention for each grid is calculated as in Equation 13.

$$A_g = Softmax \left( \frac{Q_g K_g^T}{\sqrt{D}} \right) \tag{13}$$

where $Q_g = X_g W_Q$, $K_g = X_g W_K$, $V_g = X_g W_V$. Finally, we calculate the output of each region as in Equation 14.

$$Z_g = A_g V_g \tag{14}$$

The final output is obtained as in Equation 15 by fusing the results of overlapping block attention, grid attention, and the transfer block.

$$Z_F = Concatenate(Z_{O+} + Z_{G1} + X_{out}) \tag{15}$$

$Z_O$, $Z_{G1}$ are the attention mechanism outputs from the overlapping block and grid attention, respectively, and $X_{out}$ is the transfer block.

The feature map is divided into equally sized overlapping blocks to preserve local contextual information, as shown in Fig 3. Next, self-attention is calculated within each block to model local spatial dependencies. Finally, the original spatial layout of the feature map is reconstructed using reverse partitioning. On the reconstructed feature map, an FFN is applied to process each token independently, enabling nonlinear feature transformations. Moreover, the feature map is divided into interleaved grid partitions, with attention computed across each grid to capture global contextual information. In this way, we integrate local and global contextual information from colorectal cancer images to enhance model robustness.

 

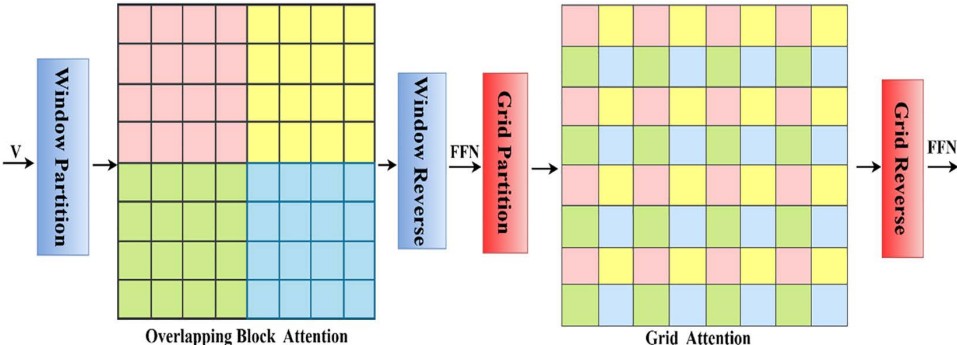

**Fig 3. Illustration of overlapping block attention and grid attention.**

### 3.5. Classification head

After the ViT encoder, we applied a classification module, which consists of a fully connected layer of 256 neurons. A dropout layer with a dropout rate of 20% is used to prevent overfitting. In addition, we applied a Softmax activation function, which gives a probability distribution over the classes, with each value being the probability of the input belonging to a specific class, as shown in equation 16.

$$Z_{Final} = Softmax(Z_F) \tag{16}$$

The loss of the model is calculated using categorical cross-entropy for multiclass classification of the colorectal diseases. This computes the negative log probability of the true class given the predicted probabilities, while penalizing incorrect predictions. For a group of $C$ classes, where the true probability of class $i$ is $p_i$, and $\hat{p}_i$ is the predicted probability of class $i$, the loss $L$ is calculated as in Equation 17.

$$L = -\sum_{i=1}^{C} p_i Log(\hat{p}_i) \tag{17}$$

The algorithm of the proposed method is as follows.

```
Algorithm 1: RNTNet for Colorectal Cancer Classification
Input: Image X ∈ R^{H×W×C}, where H is height, W is width, and C is the number of channels.
 1) Resize the input image to 300×300 pixels and normalize pixel values for model compatibility.
 2) First, reduce the number of channels by performing a 1x1 convolution using the formula:
```

$$X_1 = Conv\,1 \times 1(X)$$

```
 3) Apply a 3x3 grouped convolution to identify the image's spatial features using:
```

$$X_2 = Conv\,3 \times 3\,(X_1)$$

```
 4) Once significant features have been extracted, apply a second 1x1 convolution to expand the
    channels and enlarge the feature map as:
```

$$X_3 = Conv\,1 \times 1\,(X_2)$$

5) Apply a skip connection by performing a convolution on the input as below, where $w$ are weights and $b$ is the bias:

$$Skp = Conv(w * X + b)$$

6) Apply a ReLU activation function after combining the convolution output $X_3$ with the skip connection $Skp$ to ensure that the network can learn the identity mapping as:

$$X' = ReLU(X_3 + Skp)$$

7) The ResNeXt50 model involves stacking several blocks with different numbers of filters in stages. The multi-dimensional tensor X is flattened into a 1D vector $Y$ as follows:

$$Y = Flatten(X)$$

8) The output $Y$ is reshaped into $N$ number of patches of size $P_H \times P_W \times C$ as below:

$$X = Reshape(Y, (N, P_H, P_W, C))$$

Where $P_H, P_W, C$ are the height, width, and channels of the patches, respectively.

9) Then, the Transformer block is used to capture long-range dependencies in the input data. The attention scores are determined for each patch using:

$$Atten(Q, K, V) = softmax\left(\frac{QK^T}{\sqrt{d_k}}\right)$$

Where $Q$ is a query, $K$ is the key, $V$ is the value, $d_k$ is the dimension of key values, and $T$ is the transpose operator. The final output $X_{out}$ is calculated using $X$, $Atten$, and $FFN$, a feed-forward network followed by $Norm$, layer normalization as below:

$$X_{out} = Norm\ (X + Atten(Q, K, V) + FFN(X))$$

10) Further, for more focused attention, the image is processed using overlapping blocks by the Overlapping Block Attention module as below:

$$Z_O = \frac{\Sigma_b W_b\ (i,j) Z_b\ (i,j)}{\Sigma_b W_b\ (i,j)}$$

where $b$ represents the index of a block, $Z_b\ (i,j)$ is used to combine the weight outputs from multiple overlapping blocks, $W_b\ (i,j)$ are the weights associated with a block at $(i, j)$ position. The sum of weights is represented by $\Sigma_b W_b\ (i,j)$.

11) Also, Grid Attention is used for downsampling and self-attention on grid regions of the input as follows:

$$Z_g = A_g V_g$$

Where $A_g$ is the self-attention for each grid and $V_g$ is the value matrix for the grid region.

12) Finally, concatenate the transformer block, grid attention, and overlapping block attention outputs as below:

$$Z_F = Concatenate(X_{out} + Z_g + Z_O)$$

13) Apply a fully connected layer with 256 neurons and ReLU activation. To prevent overfitting, a dropout layer with a 20% dropout is used:

$$Dense(Z_F)$$

14) Apply the SoftMax activation function to calculate a probability distribution over the classes as follows:

$$Z_{Final} = Softmax(Z_F)$$

15) The categorical cross-entropy loss function is used to compare the predicted probabilities with the true labels as below, where the true probability of class $i$ is $p_i$, and $\hat{p}_i$ is the predicted probability of class $i$ and is calculated as:

$$L = -\sum_{i=1}^{C} p_i Log(\hat{p}_i)$$

**Output:** The class with the highest probability is the predicted class.

## 4. Results

In this section, two publicly available datasets are used, and their quantitative results for colorectal cancer disease are presented.

### 4.1. Dataset description

The first dataset is KvasirV1, which has 4000 images of anatomical landmarks, pathological findings, and polyp removal in the GI tract. The anatomical landmarks are the normal cecum (NOC), the normal pylorus (NOP), and the Z-line (NZL). The pathological findings are ulcerative colitis (ULC), polyps (POL), and esophagitis (ESO). Dyed and lifted polyps (DLP) and dyed resection margins (DRM) are polyp removal cases [28].

The second dataset is Kather, consisting of histological colorectal cancer of 150 x 150 pixels. It has eight distinct tissue types labelled as TUMORS, LYMPHO, STROMA, COMPLEX, DEBRIS, MUCOSAL, ADIPOSE, and HEALTHY. Each class contains 625 images. These are the most common tumour forms of colorectal cancer based on multiclass textures [29]. The Kather NCT-CRC-HE-100K dataset comprises 224 × 224-pixel hematoxylin and eosin (H&E) stained histopathology patches extracted from whole-slide images digitized at a resolution of 0.5 μm per pixel, corresponding to approximately 20 × optical magnification. The details of the datasets are shown in Table 1.

### 4.2. Experimental settings

We tested RNTNet with an NVIDIA Quadro RTX-4000 GPU that has 128 GB RAM and an 8GB dual graphics card. Moreover, Python version 3.10 was used for implementation. The ADAM optimiser with an initial learning rate of 0.0001 was employed to train the model for 180 epochs using a batch size of 64. A patch size of 16x16 was used in the ViT encoder to provide local and global attention to the feature map obtained from the ResNeXt.

### 4.3. Quantitative results

We trained our model with images of 300x300x3 resolution by randomly splitting the dataset into 80% and 20% for training and validation, respectively. Once the model was trained, the associated confusion matrices on the KvasirV1 and Kather validation datasets are shown in Fig 4.

The confusion matrix for the Kvasir V1 dataset in Fig 4a depicts 3135 true positives (TP) and 3065 true negatives (TN), while there are only 61 false positives (FP) and 58 false negatives (FN). This demonstrates that the proposed model exhibits excellent classification performance with high accuracy and low misclassification rates. The model can successfully distinguish between the different classes. Similar observations can be made by analyzing the confusion matrix for the Kather dataset in Fig 4b. Model outputs include 1954 TP and 1916 TN instances, while there are only 44 FN and 42 FP cases. This supports the claim that the proposed model can differentiate between various tissue types with high accuracy.

**Table 1. Summary of the datasets.**

| Datasets Used | Class Name | Type of Images | Number of Images |
|---|---|---|---|
| KvasirV1 | NOP | Endoscopic Images | 4000 |
| | NOZ | | |
| | NZL | | |
| | ULC | | |
| | POL | | |
| | ESO | | |
| | DLP | | |
| | DRM | | |
| Kather | TUMORS | Histopathology Images | 5000 |
| | LYMPHO | | |
| | STROMA | | |
| | COMPLEX | | |
| | DEBRIS | | |
| | MUCOSAL | | |
| | ADIPOSE | | |
| | HEALTHY | | |

Based on the results of the confusion matrices for both datasets, a variety of metrics were computed, including overall accuracy, precision, recall, F1-score, and Kappa, as shown in Table 2. The KvasirV1 dataset has an accuracy of 97.96%, precision of 97.96%, recall of 97.97%, and F1-score of 97.95%. Moreover, RNTNet achieved accuracy, precision, recall, and F1-score of 98.20%, 98.20%, 98.21%, and 98.20%, respectively, on the Kather dataset. The model was also tested on kappa values, which indicate that on Kather it scored a Kappa of 97.90%, while on KvasirV1 a Kappa of 97.70%.

## 5. Discussion

Conventional diagnostic methods, such as stool tests and colonoscopy, have limitations regarding early-stage sensitivity and accuracy. DL and ML strategies have been increasingly employed in automating and enhancing diagnostic precision in a bid to circumvent these challenges. The proposed RNTNet model integrates the benefits of ResNeXt, ViT, and attention mechanisms to improve feature representation and classification performance. RNTNet effectively detects colorectal abnormalities in endoscopic and histological images by integrating the extraction of spatial and contextual information with grid attention and overlapping block attention. The model was developed and tested on two clinically validated datasets, i.e., Kather, which consists of histopathological images of eight tissue classes, and KvasirV1, which consists of 4,000 endoscopic images of real-world gastrointestinal disorders.

Table 3 provides an overview of model architectures, datasets, and classification performance, focusing on accuracy, of previously published DL approaches in the diagnosis and detection of colorectal cancer. Tsai et al. [30] applied CNN on histopathology images and reported reliable tissue classification performance with an accuracy of 94.3%. Raseena et al. [31] proposed the ViT-based DeepCPD model, which achieved an excellent accuracy of 98.05%, demonstrating its efficiency in real-time colorectal polyp detection. Elshamy et al. [32] also applied a CNN in analysing histology images in their study and achieved a comparable accuracy of 98%, affirming the capability of a standard CNN when suitably optimised. However, the parallel depth-wise separable CNN (PD-CNN) of Ahamed et al. [33], which classifies images of the gastrointestinal tract, obtained a lower accuracy of 87.75%, which could be due to the complexity of the database and the variety of disease types. Raju et al. [34] designed a hybrid model using CNN and ViT for colorectal disease diagnosis and achieved an accuracy of 90.50%. The effectiveness of CNN in histological analysis was further demonstrated by Prezja et

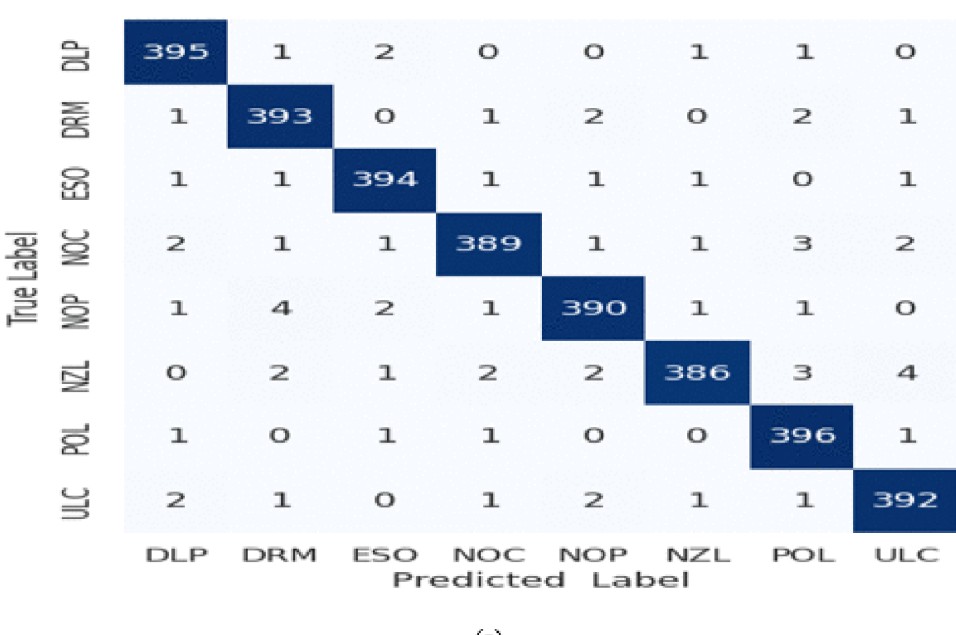

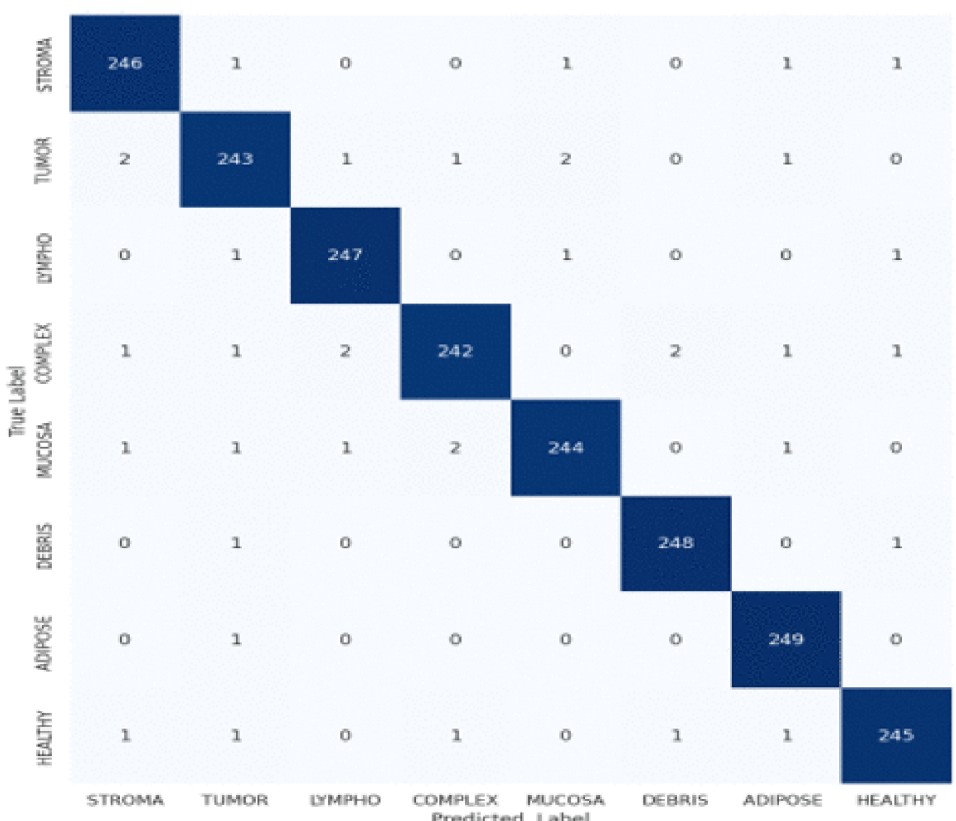

**Fig 4. Confusion matrices for (a) KvasirV1 and (b) Kather datasets.**

**Table 2. Classification performance metrics.**

| Dataset | Precision | Recall | F1-score | Accuracy | Kappa |
|---------|-----------|--------|----------|----------|-------|
| KvasirV1 | 97.96% | 97.97% | 97.95% | 97.96% | 97.70% |
| Kather | 98.20% | 98.21% | 98.20% | 98.20% | 97.90% |

**Table 3. Performance comparison with published works.**

| Author | Model | Dataset | Accuracy |
|--------|-------|---------|----------|
| Tsai et al. [30] | CNN | Histopathological Images | 94.3% |
| Raseena et al. [31] | Deep CPD (Colorectal Polyp Detection) | Colonoscopy Images | 98.05% |
| Elshamy et al. [32] | CNN | Histology Images | 98% |
| Ahamed et al. [33] | PD-CNN (Parallel Depth-wise Separable Convolutional Neural Network) | Gastrointestinal Images | 87.75% |
| Raju et al. [34] | RDV-22+BiLSTM+ViT | Colonoscopy Images | 90.50% |
| Prezja et al. [35] | CNN | Hematoxylin and eosin-stained biopsy slides | 95.6% |
| Proposed | RNTNet | Endoscopic Images & Histopathology Images | 97.96% **98.20%** |

al. [35], who used haematoxylin and eosin-stained biopsy slides and achieved an accuracy of 95.6%. With an accuracy of 97.96% and 98.20%, respectively, the suggested model, RNTNet, provided strong performance on both endoscopic and histopathological images. This implies that RNTNet surpasses most conventional CNN-based models and may provide a fair and efficient method across diverse imaging modalities.

## 5.1. Comparison with state-of-the-art on the KvasirV1 dataset

In this section, we examine the results presented in Table 4, which compares the performance of various deep learning models, i.e., InceptionV4 [36], MobileNetV4 [37], YOLOV9 [38], ResNeXt [39], MaxViT [40], and LF-ViT [41], on the KvasirV1 dataset. The results reveal a clear hierarchy among the models, with RNTNet demonstrating superior performance across all metrics, specifically, accuracy of 97.96%, precision of 97.96%, recall of 97.97%, F1-score of 97.95%, and Kappa of 97.70%. In contrast, models such as MobileNetV4 exhibit the lowest overall scores (e.g., accuracy at 88.78%), which may reflect its emphasis on computational efficiency over raw predictive power, making it more suitable for deployment in resource-limited settings. Notably, InceptionV4 shows a significant drop in recall (81.26%), indicating a tendency to miss TP instances, which is of particular importance in clinical applications, where missing abnormalities could have serious implications. ResNeXt and YOLOV9 offer balanced results, with ResNeXt's high recall (93.05%) highlighting its strength in comprehensive detection. Transformer-inspired models, i.e., MaxViT and LF-ViT, perform strongly across all metrics, underscoring the benefits of capturing both local and global contextual information in complex datasets.

## 5.2. Comparison with the state-of-the-art on the Kather dataset

Similarly, Table 5 presents the comparative performance results between state-of-the-art models on the Kather dataset, namely InceptionV4 [36], MobileNetV4 [37], YOLOV9 [38], ResNeXt [39], MaxViT [40], and LF-ViT [41]. A distinct performance gradient emerges with the proposed model leading in every metric. Specifically, it achieves an accuracy of 98.20%, precision of 98.21%, recall of 98.20%, F1-score of 98.20%, and Kappa of 97.90%. Conversely, MobileNetV4 trails with an accuracy of 90.87%, likely due to its streamlined architecture favoring speed and low resource demands over peak discriminative power. Mid-range models, YOLOV9 and ResNeXt, deliver consistent results, i.e., accuracies of 92.15% and 94.72%, respectively, with ResNeXt notable for providing consistent performance across all metrics, supporting its use on applications requiring steady detection rates. MaxViT and LF-ViT, achieve excellent performance with F1-scores

**Table 4. Results comparison on the KvasirV1 dataset.**

| Model | Accuracy(%) | Precision(%) | Recall(%) | F1-score(%) | Kappa(%) |
|---|---|---|---|---|---|
| InceptionV4 | 92.46 | 90.08 | 81.26 | 85.44 | 91.12 |
| MobileNetV4 | 88.78 | 87.60 | 88.18 | 87.89 | 87.89 |
| YOLOV9 | 91.25 | 89.87 | 90.40 | 90.13 | 90.38 |
| ResNeXt | 93.21 | 92.54 | 93.05 | 92.79 | 92.72 |
| MaxViT | 95.16 | 94.80 | 93.48 | 94.13 | 94.29 |
| LF-ViT | 96.08 | 95.70 | 94.98 | 95.34 | 95.27 |
| Proposed | **97.96** | **97.96** | **97.97** | **97.95** | **97.70** |

**Table 5. Results comparison on the Kather dataset.**

| Model | Accuracy(%) | Precision(%) | Recall(%) | F1-score(%) | Kappa(%) |
|---|---|---|---|---|---|
| InceptionV4 | 94.07 | 93.12 | 94.61 | 93.86 | 93.85 |
| MobileNetV4 | 90.87 | 89.16 | 88.70 | 88.93 | 90.02 |
| YOLOV9 | 92.15 | 90.32 | 91.49 | 90.90 | 91.53 |
| ResNeXt | 94.72 | 93.04 | 92.94 | 92.99 | 93.08 |
| MaxViT | 96.13 | 95.40 | 94.69 | 95.04 | 95.25 |
| LF-ViT | 96.94 | 96.24 | 95.19 | 95.71 | 95.57 |
| Proposed | **98.20** | **98.21** | **98.20%** | **98.20** | **97.90** |

of 95.04% and 95.71%, respectively, demonstrating their efficacy in managing intricate spatial relationships within the dataset.

### 5.3. ROC analysis

In Fig 5a, the proposed model works best in the KvasirV1 dataset with the top AUC of 0.9895. LF-ViT achieved an AUC of 0.9738, and MaxViT had an AUC of 0.9630. ResNeXt, InceptionV4, and YOLOV9 have AUCs of 0.9417, 0.9326, and 0.9216, respectively. MobileNetV4 has the lowest AUC of 0.9017. Similarly, RNTNet achieved an AUC of 0.9937 on the Kather dataset, as shown in Fig 5b, which is the highest compared to other models. MaxViT and LF-ViT have AUC values of 0.9750 and 0.9806, respectively. Other models, YOLOV9, ResNeXt, and InceptionV4, have AUCs of 0.9345, 0.9504, and 0.9512, respectively. MobileNetV4 exhibits the lowest AUC of 0.9207.

### 5.4. Training stability

The loss curves shown in Fig 6 show the different convergence patterns for the proposed model trained on the KvasirV1 and Kather datasets over 180 epochs. During the first 25 epochs, the training loss for the KvasirV1 dataset drops to less than 0.1 from a starting value of about 0.5. After 180 epochs, the training loss stabilizes at about 0.02. Similarly, the validation loss begins at about 0.8 and stabilizes at about 0.03 at 180 epochs. The Kather dataset shows an initial loss of 0.4 and drops to 0.05 over the following 25 epochs, and at 180 epochs, the training loss is 0.01. However, the validation loss begins at 0.7 and is reduced to 0.02 at nearly 180 epochs. It could be observed from the loss curves that the proposed model has achieved low training and validation losses, indicating better learning stability.

### 5.5. Ablation study

Table 6 provides the ablation analysis on the integration of the various attention mechanisms into the hybrid model combining ResNeXt and ViT on the KvasirV1 dataset. The baseline models, ResNeXt and ViT, individually, are found

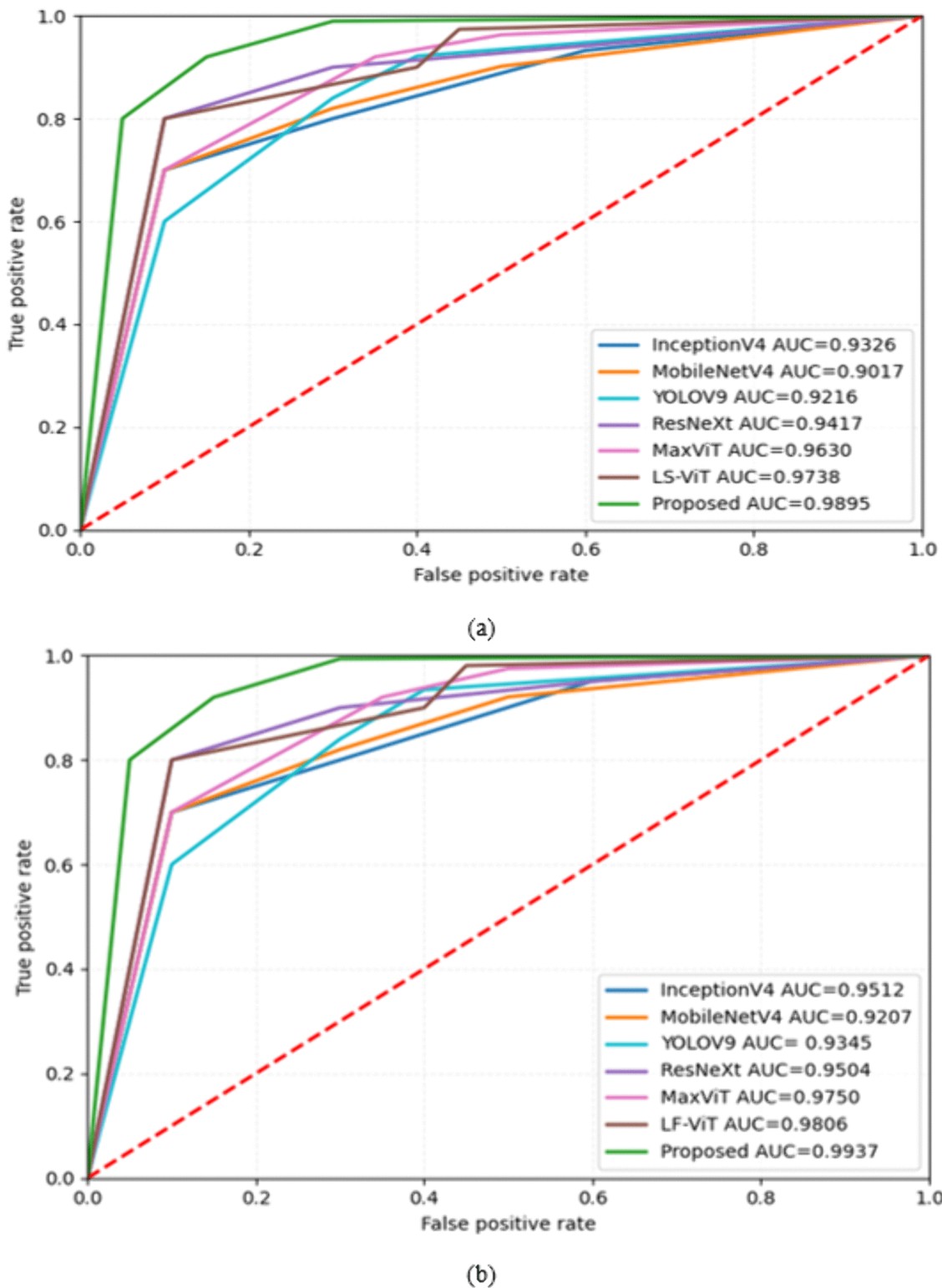

**Fig 5. ROC curves for (a) KvasirV1 (b) Kather datasets.**

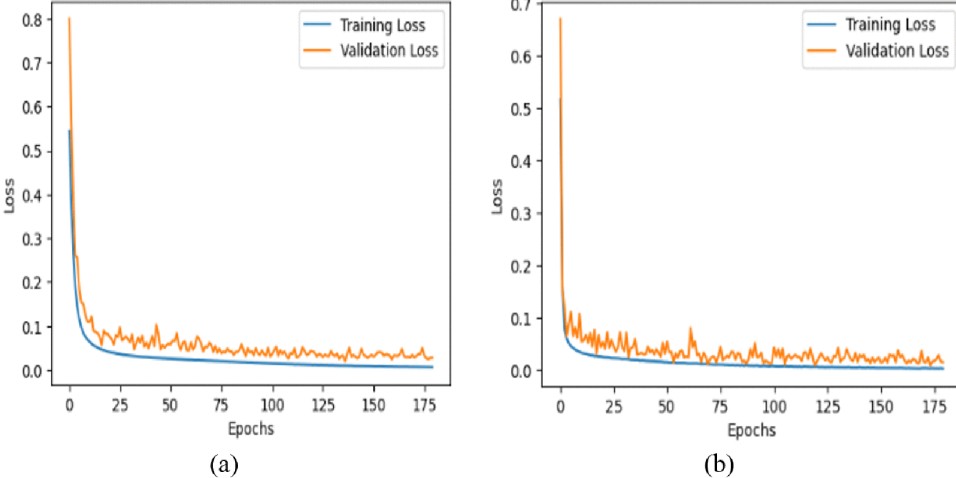

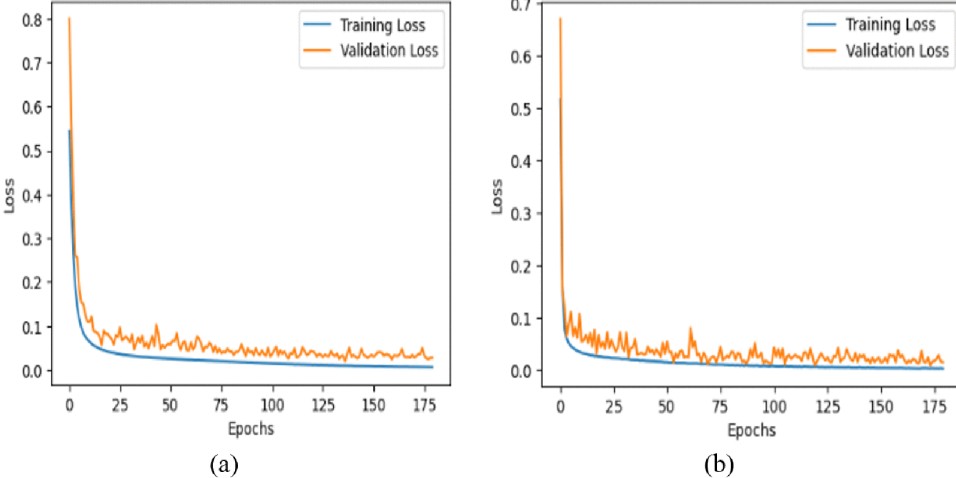

**Fig 6. Loss curves of the (a) KvasirV1 and (b) Kather dataset.**

**Table 6. Ablation study using different components.**

| Dataset | Components | Precision | Kappa |
|---|---|---|---|
| KvasirV1 | ResNeXt | 93.19% | 92.72% |
| | ViT | 94.67% | 93.50% |
| | ResNeXt+ViT (MHSA) | 96.13% | 95.94% |
| | ResNeXt+ViT (OBA) | 95.15% | 95.06% |
| | ResNeXt+ViT (GA) | 96.07% | 95.80% |
| | ResNeXt+ViT (OBA+GA) | 97.96% | 97.70% |
| Kather | ResNeXt | 94.28% | 94.03% |
| | ViT | 95.14% | 94.89% |
| | ResNeXt+ViT (MHSA) | 97.05% | 96.49% |
| | ResNeXt+ViT (OBA) | 96.25% | 95.48% |
| | ResNeXt+ViT (GA) | 97.10% | 96.37% |
| | ResNeXt+ViT (OBA+GA) | 98.20% | 97.90% |

to produce precisions of 93.19% and 94.67%, respectively. When the combination of these models is used with MHSA, performance improved with a precision of 96.13% and Kappa value of 95.94%. Additional experiments used alternative attention mechahisms, including OBA and GA. ResNeXt+Vi (OBA) achieves a precision of 95.15%, whereas ResNeXt+ViT (GA) obtains a precision of 96.07%, and thus, it is concluded that GA is more beneficial compared to OBA. When both OBA and GA are used together with the ResNeXt+ViT structure, the system achieves an optimal performance with a precision value of 97.96% and a Kappa value of 97.70%.

Similarly, the ablation experiment on the Kather dataset examines the integration of various architectural elements and attention mechanisms in the hybrid deep learning model consisting of ResNeXt and Vision Transformer. ResNeXt has a precision of 94.28% and Kappa of 94.03%, whereas ViT does better with a precision of 95.14% and a Kappa of 94.89%. This comparison implies that transformer models have the edge in extracting long-range dependencies from the data. When both architectures are used together with MHSA, model performance improves substantially to a precision of 97.05% and a Kappa of 96.49%. Including OBA in the hybrid model results in a slight increase in precision and Kappa

of 96.25% and 95.48%, respectively. Alternatively, adding GA increases the performance further to a precision of 97.10% and a Kappa of 96.37%, supporting its suitability in capturing global contextual information.

In summary, the ablation experiments on the two datasets show that the integration of convolutional and transformer models with attention significantly improves performance. Performance on both KvasirV1 and Kather demonstrates more improvement when OBA and GA are used together, while when it comes to using GA alone, it is evident that this improves performance more than using OBA alone.

**5.5.1. Complexity analysis.** Table 7 provides an evaluation of the computational efficiency of various deep learning architectures in the context of colorectal disease classification. In order to showcase the trade-off between performance and resource requirements, we use training time (in minutes), inference time (in seconds), floating-point operations (GFLOPs), and model parameters (in millions). RNTNet demonstrates a balanced profile with a training duration of 123 minutes and inference speed of 92 seconds, coupled with 5.12 GFLOPs and 20.82 million parameters, positioning it as an efficient system that outperforms heavier models, e.g., YOLOV9 (273 minutes training, 135 seconds inference, 189 GFLOPs, and 57.3 million parameters) in terms of reduced computational overhead, while maintaining competitive efficacy, attributable to its optimized attention mechanisms and lightweight feature extraction akin to those in LF-ViT (125 minutes training, 81 seconds inference, 1.90 GFLOPs, and 22 million parameters). In contrast, models such as InceptionV4 and MaxViT exhibit higher complexity. For instance, InceptionV4 requires 167 minutes for training and 13 GFLOPs with 44 million parameters, whereas MaxViT demands 141 minutes of training and 23.4 GFLOPs alongside a substantial 119 million parameters, being more suitable in resource-rich environments, but potentially limited for deployment on edge devices or real-time diagnostics. Lightweight options, e.g., MobileNetV4 offer the lowest parameter count at 11.15 million and 5.62 GFLOPs, with 105-minute training and 74-second inference, yet they compromise accuracy for speed, while ResNeXt strikes a middle ground with 113 minutes training, 87 seconds inference, 4.20 GFLOPs, and 25 million parameters, underscoring its efficiency for scalable applications. Overall, the proposed model offers enhanced practicality in clinical settings, where computational constraints could favor models that minimize training overhead and parameter overheads without sacrificing diagnostic robustness, paving the way for further optimizations in hybrid architectures to balance these factors more effectively.

**5.5.2. Model decision analysis using Grad-CAM.** The model decision process is essential for understanding which regions of interest are used in the classification tasks. We used Grad-CAM, a powerful visualization technique, to enhance the interpretability of RNTNet on CRC, to provide potential end-users with insights into which regions of the colorectal images most influence the model's predictions. Moreover, medical experts could use the Grad-CAM results to obtain a second opinion. Fig 7 visualizes the Grad-CAM results for the Complex, Mucosa, Tumor, and Stroma classes of the Kather dataset. The model focuses on the regions of complex cellular arrangements observed in Fig 7a, which belong to the Complex class and indicate abnormal tissue structures. As indicated in Fig 7b, the Mucosa class features high activation around mucosal linings, which are significant indicators of colorectal disease. The model confirms its ability to identify tumor characteristics by appropriately distinguishing and indicating areas expected to foster cancerous cells in Fig 7c, which

**Table 7. Complexity comparison with SOTA methods.**

| Model | Train(m) | Inference(s) | GFlops | Parameters(M) |
|---|---|---|---|---|
| InceptionV4 | 167 | 83 | 13 | 44 |
| MobileNetV4 | 105 | 74 | 5.62 | 11.15 |
| YOLOV9 | 273 | 135 | 189 | 57.3 |
| ResNeXt | 113 | 87 | 4.20 | 25 |
| MaxViT | 141 | 93 | 23.4 | 119 |
| LF-ViT | 125 | 81 | 1.90 | 22 |
| Proposed | 123 | 92 | 5.12 | 20.82 |

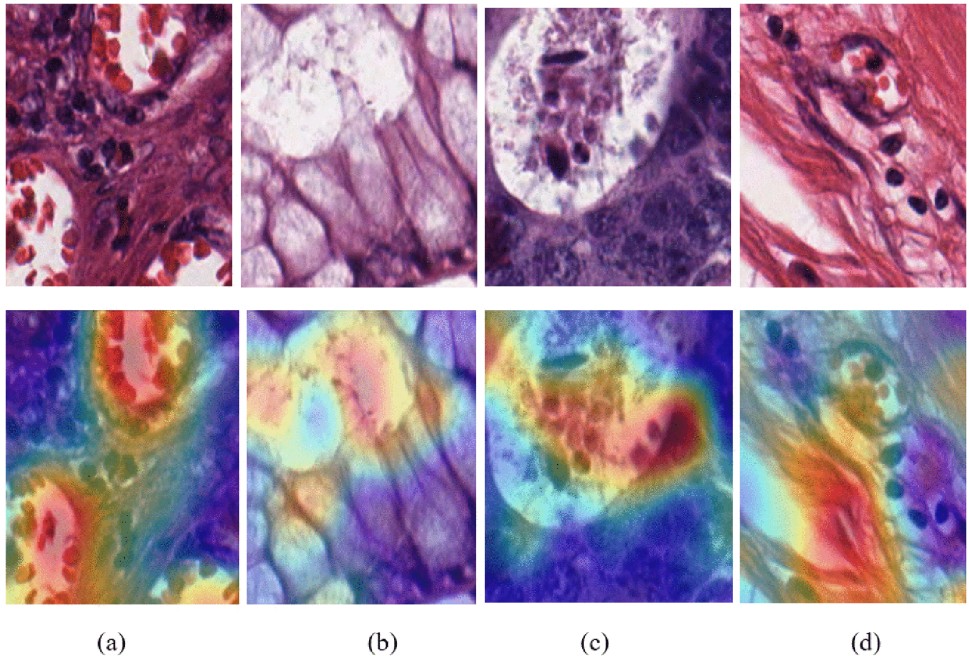

| (a) | (b) | (c) | (d) |

**Fig 7. Decision process analysis using Grad-CAM: (a) Complex (b) Mucosa (c) Tumor, and (d) Stroma classes of the Kather dataset.**

correspond to the Tumor class. Lastly, the Stroma class, shown in Fig 7d, indicates fibrous connective tissue areas often involved in tumor microenvironments. Apart from having good classification results, these Grad-CAM outputs demonstrate that RNTNet concentrates on medically relevant details, making its predictions understandable and useful for clinical decision-making.

**5.5.3. Effect of patch size.** This section presents a comparison of the RNTNet model's accuracy on two datasets, Kather and KvasirV1, with varying patch sizes, i.e., 4×4, 8×8, 16×16, 64×64, and 128×128, as shown in Fig 8. With a precision of 95.17% on the Kather dataset and 94.27% on the KvasirV1 dataset, the model performed poorly at the smallest patch size of 4×4, which implies that there was insufficient contextual information. RNTNet achieved a precision of 96.05% on Kather and 95.87% on KvasirV1, when the patch size was increased to 8×8, while the highest precision of 98.20% for Kather and 97.96% for KvasirV1, was obtained with a patch size of 16×16. Precision dropped when the patch size changed to 64×64. Furthermore, with a patch size of 128×128, precision dropped further to 96.89% and 95.78%, on Kather and KvasirV1, respectively. These results indicate that the best patch size for achieving the highest classification performance for colorectal cancer detection is 16×16, while patch sizes that are both too small and too large negatively affect the model performance.

**5.5.4. Statistical analysis.** We performed statistical analysis, specifically, a paired t-test, on the performance results presented in Table 6, with the outcomes presented in Figs 9 and 10. We employed the paired t-test for per-class performance indicators (precision and Cohen's kappa) on the eight classes of the KvasirV1 and Kather datasets. The model's predicted outcomes for every class were compared to the ground truth to obtain the per-class metric values. These values were then paired and analyzed using a paired t-test to determine if the differences between measures (such as accuracy vs. kappa) were statistically significant. This approach was selected to ensure that the statistical test reflected both class-wise variability and aggregate performance.

Fig 9 shows that the mean difference between precision and Kappa value on the KvasirV1 dataset is 0.44. Further, the p-value is 0.0381, which is much smaller than the statistically significant value of 0.05. Hence, the null hypothesis

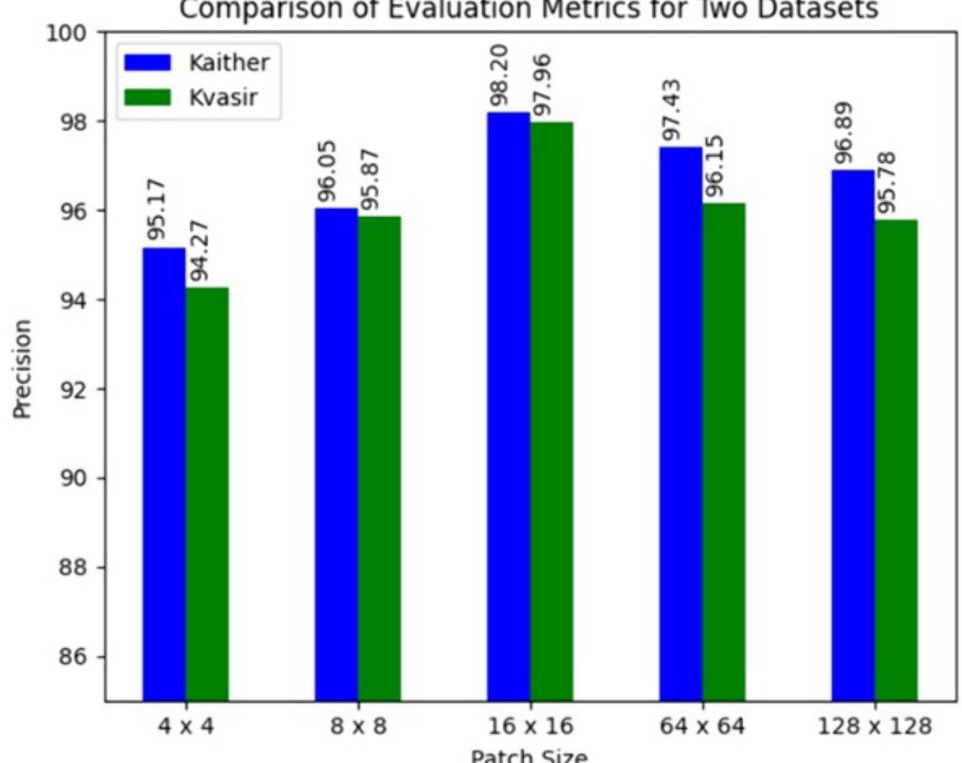

**Fig 8. Effect of patch size on Kather and KvasirV1 colorectal cancer datasets.**

can be rejected. Moreover, Fig 10 shows a mean difference value of 0.48 on the Kather dataset. Further, the p-value of $0.00472 < 0.05$ confirms its statistically significant nature, and the null hypothesis can be rejected.

**5.5.5. Performance evaluation using 5-fold cross-validation.** We applied a 5-fold cross-validation scheme to train and test the model's performance on the Kather and KvasirV1 datasets. The quantitative results are depicted in Tables 8 and 9.

Table 8 displays the performance of the proposed model on the KvasirV1 dataset using 5-fold cross-validation. In Fold 1, the model showed consistent initial performance, attaining 95.18% accuracy with a precision of 94.30%, a recall of 93.89%, and an F1-score of 94.07%. Fold 2 demonstrates an accuracy of 96.40%, a recall of 94.29%, and a precision of 95.06%. Fold 3 further enhanced performance with a 96.87% accuracy. Fold 4 achieved an accuracy of 97.08% and an F1-score of 96.41%. The highest performance is observed in Fold5, with an accuracy of 98.67%, F1-score of 97.40% and Kappa of 97.14%. Overall, the average performance of all five folds is 96.80% accuracy, 95.90% precision, 95.43% recall, 95.57% F1-score, and 95.29% kappa. The results show that the proposed method performs consistently and reduces the risk of overfitting to the KvasirV1 dataset.

The performance of the proposed model on the Kather dataset across all five cross-validation folds is shown in Table 9. The model's accuracy in Fold 1 was 96.06% with 95.81% precision. Fold 2 showed improvement with 97.43% accuracy, 96.28% precision, and 95.86%. Further, the Fold 3 model achieved an accuracy and precision value of 98.13% and 97.09%, respectively. Fold 4 showed steady generalization with 98.50% accuracy, 97.52% precision, 97.12% recall, 97.37% F1-score, and 97.03% Kappa. The model obtained 97.49% Kappa, 98.17% recall in Fold 5. Moreover, the model's average precision and Kappa score are 96.98% and 96.17%, respectively.

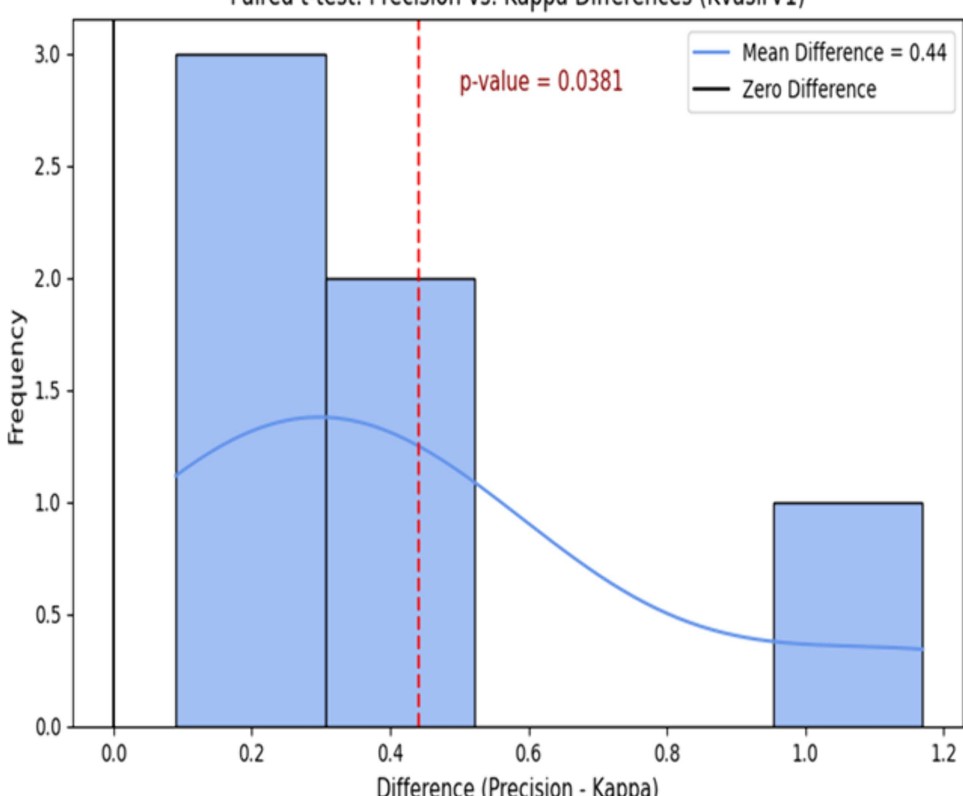

**Fig 9. Paired t-test on the KvasirV1 dataset.**

## 6. Conclusion

In this paper, we propose RNTNet using a residual network and a ViT encoder with two attention mechanisms, i.e., overlapping block attention and grid attention, in the classification of colorectal cancer images. ResNeXt is capable of providing high-dimensional spatial features from the CRC images. At the same time, the mixed attention mechanism provides multi-scale global attention to the feature map, which helps to improve the diagnosis results by focusing on the complex patterns of the diseases. The KvasirV1 dataset contains endoscopic images, on which our model obtained accuracy and recall values of 97.96% and 97.97%, respectively. Moreover, it obtained an AUC of 0.9895, as ascertained by ROC analysis. At the same time, on the Kather dataset, which contains histopathological images, RNTNet had accuracy and Kappa values of 98.20% and 97.90%, respectively. The training loss and the validation loss converged to ~0.02 on KvasirV1 and ~0.01 on Kather at 180 epochs, signifying good optimization and low overfitting. Moreover, Grad-CAM results suggest that the model focuses on regions of interest that oncologists can use to obtain a second opinion. The ablation study highlighted the contributions of each component and showed that the combination of OBA and GA with ResNeXt and ViT produced the best results. Our results are based solely on benchmark datasets, and before any claims of diagnostic value can be made, clinical-level validation on patient cohorts, including outcomes and metadata, would be necessary. The computational costs of the mixed attention require reduction through computationally efficient implementations of attention mechanisms. In addition, the features extracted through the ResNeXt block can be further optimized by using nature-inspired optimization algorithms.

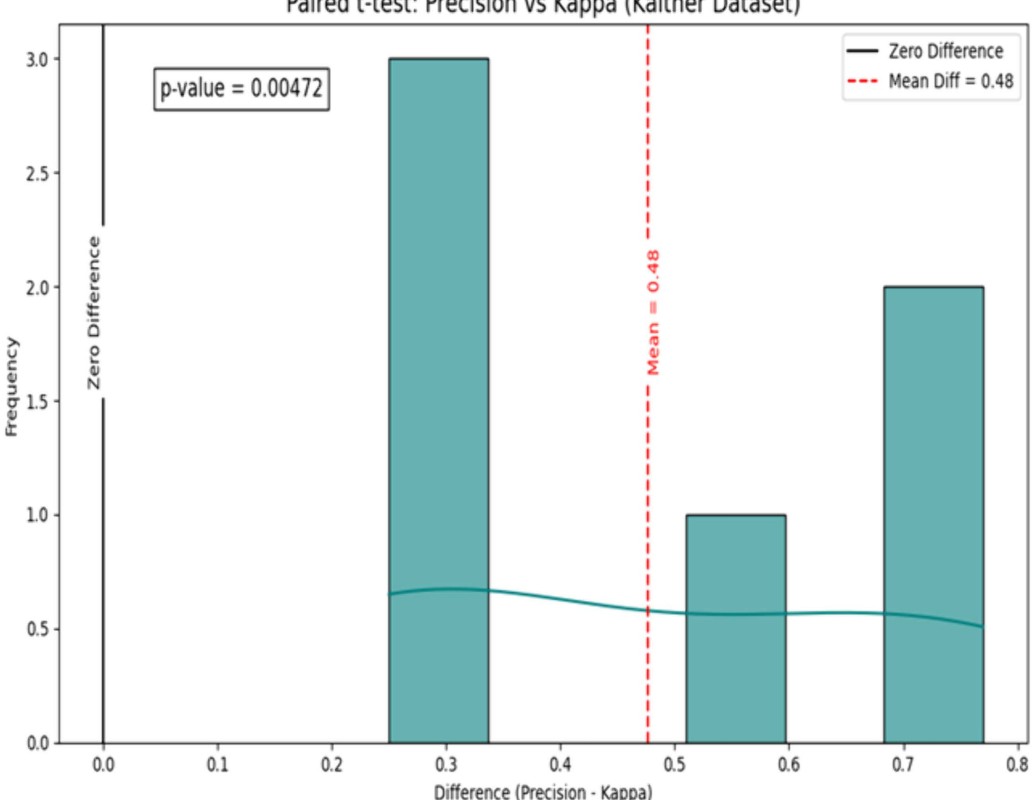

**Fig 10. Paired t-test on the Kather dataset.**

**Table 8. Performance of the proposed model on the KvasirV1 dataset using 5-fold cross-validation.**

| Folds | Accuracy (%) | Precision (%) | Recall (%) | F1-score (%) | Kappa (%) |
|---|---|---|---|---|---|
| Fold_1 | 95.18 | 94.30 | 93.89 | 94.07 | 93.48 |
| Fold_2 | 96.40 | 95.06 | 94.29 | 94.97 | 94.16 |
| Fold_3 | 96.87 | 96.10 | 95.60 | 95.90 | 95.48 |
| Fold_4 | 97.08 | 96.57 | 96.08 | 96.41 | 96.17 |
| Fold_5 | 98.67 | 97.46 | 97.30 | 97.40 | 97.14 |
| **Average** | **96.80** | **95.90** | **95.43** | **95.75** | **95.29** |

**Table 9. Performance of the proposed model on the Kather dataset using 5-fold cross-validation.**

| Folds | Accuracy (%) | Precision (%) | Recall (%) | F1-score (%) | Kappa(%) |
|---|---|---|---|---|---|
| Fold_1 | 96.06 | 95.81 | 94.79 | 95.33 | 94.31 |
| Fold_2 | 97.43 | 96.28 | 95.86 | 96.22 | 95.45 |
| Fold_3 | 98.13 | 97.09 | 96.73 | 96.91 | 96.54 |
| Fold_4 | 98.50 | 97.52 | 97.12 | 97.37 | 97.03 |
| Fold_5 | 98.67 | 98.20 | 98.17 | 98.11 | 97.49 |
| **Average** | **97.76** | **96.98** | **96.53** | **96.79** | **96.17** |

## Author contributions

**Conceptualization:** Poonam Sharma, Bhisham Sharma, Dhirendra Prasad Yadav.

**Data curation:** Bhisham Sharma, Dhirendra Prasad Yadav.

**Formal analysis:** Ajit Noonia, Panos Liatsis.

**Investigation:** Bhisham Sharma.

**Methodology:** Poonam Sharma, Bhisham Sharma, Dhirendra Prasad Yadav.

**Project administration:** Ajit Noonia, Panos Liatsis.

**Resources:** Panos Liatsis.

**Software:** Bhisham Sharma.

**Visualization:** Poonam Sharma, Bhisham Sharma.

**Writing – original draft:** Poonam Sharma, Dhirendra Prasad Yadav.

**Writing – review & editing:** Bhisham Sharma, Ajit Noonia, Panos Liatsis.

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
