## [Decision Letter · Decision Letter 0]

21 Jul 2025

Dear Dr. Liatsis,

Thank you for submitting your manuscript to PLOS ONE. After careful consideration, we feel that it has merit but does not fully meet PLOS ONE’s publication criteria as it currently stands. Therefore, we invite you to submit a revised version of the manuscript that addresses the points raised during the review process.

While the reviewers find your work to be of interest, they have identified several important issues that must be addressed through a major revision. We therefore invite you to revise your manuscript, carefully considering the reviewers’ comments and adhering to the general guidelines provided below.

To facilitate a smooth peer review process and help the Editor and Reviewers make a final recommendation, please ensure that you respond thoroughly to each of the comments outlined at the end of this message. A detailed and thoughtful revision will help avoid further rounds of clarification and expedite the editorial decision.

We look forward to receiving your revised manuscript.

Kind regards,

Siamak Pedrammehr,

Ph.D.

Academic Editor

PLOS ONE

Journal Requirements:

Reviewers' comments:

Reviewer's Responses to Questions

**Comments to the Author**

1. Is the manuscript technically sound, and do the data support the conclusions?

Reviewer #1: Partly

Reviewer #2: Yes

2. Has the statistical analysis been performed appropriately and rigorously?

Reviewer #1: No

Reviewer #2: Yes

3. Have the authors made all data underlying the findings in their manuscript fully available?

Reviewer #1: Yes

Reviewer #2: Yes

4. Is the manuscript presented in an intelligible fashion and written in standard English?

Reviewer #1: No

Reviewer #2: Yes

Reviewer #1: Thank you for the opportunity to review your manuscript entitled “Early diagnosis of colorectal cancer using residual transformer with mixed attention and explainable AI.” The topic is timely and important, and your attempt to enhance deep learning-based histopathological classification through transformer models and explainable AI is commendable. However, after careful evaluation, I regret to recommend rejection of the manuscript in its current form due to the following significant concerns:

1. Lack of Clear Novelty

While the manuscript introduces a “residual transformer with mixed attention,” it does not sufficiently differentiate this model from existing Vision Transformer (ViT) architectures. The conceptual approach, using transformers with GradCAM for histopathology classification, has already been reported in prior studies (Colon Cancer Detection using Vision Transformers and Explainable AI), including work that closely parallels the structure and aims of your paper. Without a clearly defined architectural innovation or substantial methodological advancement, the manuscript falls short of meeting the novelty threshold expected for publication.

2. Incomplete Methodological Transparency

Several key components of your methodology are inadequately described. This includes image preprocessing (e.g., patch generation, magnification level), training configuration (e.g., optimizer, learning rate, epochs), and validation design (e.g., test set construction, use of cross-validation). These omissions hinder reproducibility and make it difficult to assess the rigor of your experimental setup.

3. Insufficient Statistical Validation

The manuscript lacks statistical testing to support claims of model superiority. Metrics such as accuracy and F1-score are presented without confidence intervals or hypothesis testing. No standard deviations or cross-validation results are reported. Without statistical evidence, the performance differences between RTMA and baseline models remain unsubstantiated.

4. Overstated Clinical Claims

The manuscript suggests potential application in “early diagnosis” of colorectal cancer. However, the dataset used (LC25000) is synthetic, lacks patient-level metadata, and is not validated against real-world clinical outcomes. As such, claims about early diagnostic relevance are premature and should be avoided in the absence of clinical data or external validation.

5. Language and Presentation Issues

While the manuscript is generally understandable, there are numerous grammatical issues, repetitive phrasing, and instances of vague or informal technical language. These issues negatively impact the readability and professionalism of the work.

Reviewer #2: The model architecture is described in details and the analysis is sound. I thank the authors for this solid work. the ViT architecture is super capable in handling image data compared to previous models.

**Do you want your identity to be public for this peer review?** For information about this choice, including consent withdrawal, please see our Privacy Policy

Reviewer #1: No

Reviewer #2: **Yes: ** Hamidreza Ashayeri

---

## [Author Response · Author response to Decision Letter 1]

8 Aug 2025

Point-by-point Response to Editor and Reviewers’ Comments

Title: Early Diagnosis of Colorectal Cancer Using Residual Transformer with Mixed Attention and Explainable AI

Authors: Poonam Sharma, Bhisham Sharma, Ajit Noonia, Dhirendra Prasad Yadav, Panos Liatsis

Response to the Reviewers Comments:

Reviewer #1:

Comment 1: Lack of Clear Novelty. While the manuscript introduces a “residual transformer with mixed attention,” it does not sufficiently differentiate this model from existing Vision Transformer (ViT) architectures. The conceptual approach, using transformers with GradCAM for histopathology classification, has already been reported in prior studies (Colon Cancer Detection using Vision Transformers and Explainable AI), including work that closely parallels the structure and aims of your paper. Without a clearly defined architectural innovation or substantial methodological advancement, the manuscript falls short of meeting the novelty threshold expected for publication.

Response: Thank you for your concern related to the novelty of the proposed model. Previous efforts [11] attempted to tackle this challenge using a vision transformer model, which incorporates multi-head self-attention (MHSA) to provide for global correlations of the spatial features. However, the computational costs of the MHSA are quadratic in time, and require a high volume of training data for appropriate model generalization. Moreover, the classical transformer model lacks local spatial features; consequently, local and complex textural features are not adequately captured. Further, their model was tested on the binary classification task.

The proposed Residual Next Transformer Network (RNTNet) uses ResNeXt to extract high dimensional local spatial features from colon cancer images. Further, mixed attention-based transformer encoder is applied to the flattened features obtained from the ResNeXt block, which provides local and global attention to the feature map through the overlapping block attention and grid block attention, respectively. Due to this, the model is capable of focusing on the edge and boundary regions of the colon cancer regions. In addition, the computational costs of the mixed attention block are O (M2D2), where M is the feature map size and D is the dimension, while those of MHSA in the classical ViT are O (M4D), which is much higher than the proposed model. Further, we tested RNTNet on the diverse datasets KvasirV1 and Kaither, which contain 8 types of diseases, i.e., multi-class classification.

The contributions of this research are as follows:

(1) We utilized the ResNeXt, a group convolution-based residual network for high-dimensional spatial feature extraction and the ViT encoder to provide global contextual information for precise CRC diagnosis.

(2) Further, in the ViT encoder, we integrated overlapping block attention and grid attention to effectively capture local and global spatial and contextual information of the colorectal cancer regions.

(3) A Grad-Cam module was incorporated in the proposed system to provide explainability on the decision process of the model to support its use as a second opinion to oncologists.

(4) We used two datasets, KvasirV1 and Kaither, to evaluate the performance of the model and compared with state-of-the-art (SOTA) methods.

[11] Prabhu, A.J. (2024). Colon cancer detection using vision transformers and explainable AI. International Journal of Advanced Research in Science, Communication and Technology (IJARSCT), 4(1), 189-196.

Comment 2: Incomplete Methodological Transparency Several key components of your methodology are inadequately described. This includes image preprocessing (e.g., patch generation, magnification level), training configuration (e.g., optimizer, learning rate, epochs), and validation design (e.g., test set construction, use of cross-validation). These omissions hinder reproducibility and make it difficult to assess the rigor of your experimental setup.

Response: We thank the reviewer for highlighting the importance of reproducibility and methodological transparency. We acknowledge that certain pieces of information in our initial submission were deficient, and we have already made changes to the text to address these gaps. Specifically:

Image preprocessing and patch generation: We resized images to 300x300x3 for the input to ResNeXt. In the ViT encoder a non-overlapping patch of 16 x 16 is used for the input. Images in the Kaither dataset were histopathology slides with eight tissue classes and a resolution of 150 x 150. Since the dataset was evenly preprocessed, no adjustments to the magnification level were made.

5.5.3. Effect of patch size

This section presents a comparison of the RNTNet model's accuracy on two datasets, Kaither and KvasirV1, with varying patch sizes, i.e., 4×4, 8×8, 16×16, 64×64, and 128×128, as shown in Figure 8. With a precision of 95.17% on the Kaither dataset and 94.27% on the KvasirV1 dataset, the model performed poorly at the smallest patch size of 4×4, which implies that there was insufficient contextual information. RNTNet achieved a precision of 96.05% on Kaither and 95.87% on KvasirV1 when the patch size was increased to 8×8, while the highest precision of 98.20% for Kaither and 97.96% for KvasirV1, was obtained with a patch size of 16×16. Precision dropped when the patch size changed to 64×64. Furthermore, with a patch size of 128×128, precision dropped further to 96.89% and 95.78%, on Kaither and KvasirV1, respectively. These results indicate that the best patch size for achieving the highest classification performance for colorectal cancer detection is 16×16, while patch sizes that are both too small and too large negatively affect the model performance.

Figure 8. Effect of patch size on Kaither and Kvasir colorectal cancer datasets

Training Configuration: We provided the details of the training configuration in Section 4.2 as follows:

We tested RNTNet with an NVIDIA Quadro RTX-4000 GPU that has 128 GB RAM and an 8GB dual graphics card. Moreover, Python version 3.10 was used for implementation. The ADAM optimizer with a initial learning rate of 0.0001 was employed to train the model for 180 epochs using a batch size of 64. A patch size of 16x16 was used in the ViT encoder to provide local and global attention to the feature map obtained from the ResNeXt.

Validation Design: In section 4.3, we explicitly stated that the dataset was randomly split into 80% for training and 20% for validation. The number of images in each class is equally distributed in both classes. Therefore, we avoid the use of cross-validation to evaluate model performance.

Comment 3: Insufficient Statistical Validation. The manuscript lacks statistical testing to support claims of model superiority. Metrics such as accuracy and F1-score are presented without confidence intervals or hypothesis testing. No standard deviations or cross-validation results are reported. Without statistical evidence, the performance differences between RTMA and baseline models remain unsubstantiated.

Response: Thank you for your valuable comment. We added Section 5.5.4 Statistical Analysis as follows:

We performed statistical analysis, specifically, a paired t-test, on the performance results presented in Table 6, with the outcomes presented in Figures 9 and 10. Figure 9 shows that the mean difference between precision and Kappa value on the KvasirV1 dataset is 0.44. Further, the p-value is 0.0381, which is much smaller than the statistically significant value of 0.5. Hence, the null hypothesis can be rejected. Moreover, Figure 10 shows a mean difference value of 0.48 on the Kaither dataset. Further, the p-value of 0.00472 <0.5 confirms its statistically significant nature, and the null hypothesis can be rejected.

Figure 9. Statistical analysis on the KvasirV1 dataset

Figure 10. Statistical analysis on the Kaither dataset

Comment 4: Overstated Clinical Claims. The manuscript suggests potential application in “early diagnosis” of colorectal cancer. However, the dataset used (LC25000) is synthetic, lacks patient-level metadata, and is not validated against real-world clinical outcomes. As such, claims about early diagnostic relevance are premature and should be avoided in the absence of clinical data or external validation.

Response: Thank you for this important comment. We utilized two datasets, i.e., Kaither and KvasirV1, which are verified by experts and all annotations are provided by them for disease diagnosis.

The dataset used in the study can be accessed using URL: https://zenodo.org/records/53169

https://datasets.simula.no/kvasir/.

Comment 5: Language and Presentation Issues. While the manuscript is generally understandable, there are numerous grammatical issues, repetitive phrasing, and instances of vague or informal technical language. These issues negatively impact the readability and professionalism of the work.

Response: Thank you for your valuable suggestion. We performed a thorough review of the manuscript by a native English speaker, and dealt with all grammatical errors, eliminated repetitive phrasing, and replaced informal or unclear technical language with academically appropriate terms. The manuscript has undergone a thorough revision process.

Reviewer #2:

The model architecture is described in details and the analysis is sound. I thank the authors for this solid work. the ViT architecture is super capable in handling image data compared to previous models.

Response: Thank you for your valuable feedback.

---

## [Decision Letter · Decision Letter 1]

8 Sep 2025

Dear Dr. Liatsis, 

We look forward to receiving your revised manuscript.

Kind regards,

Siamak Pedrammehr, Ph.D.

Academic Editor

PLOS ONE

Journal Requirements:

Reviewers' comments:

Reviewer's Responses to Questions

**Comments to the Author**

Reviewer #1: (No Response)

Reviewer #2: All comments have been addressed

2. Is the manuscript technically sound, and do the data support the conclusions?

Reviewer #1: Partly

Reviewer #2: Yes

3. Has the statistical analysis been performed appropriately and rigorously?

Reviewer #1: No

Reviewer #2: Yes

4. Have the authors made all data underlying the findings in their manuscript fully available?

Reviewer #1: Yes

Reviewer #2: Yes

5. Is the manuscript presented in an intelligible fashion and written in standard English?

Reviewer #1: Yes

Reviewer #2: Yes

Reviewer #1: 1. Thank you for your responses; however, ResNeXt + ViT hybridization and mixed attention are not conceptually novel. Others have done similar hybrids. What’s the actual innovation beyond rearranging known components?

2. State the actual slide magnification (or explicitly say if that info wasn’t available). For example, “all slides were provided at 40× magnification.”

3. Trim hardware specs down to just GPU model + VRAM. The kind of detail you wrote isn’t as important for reproducibility as batch size, optimizer, etc.

4. You wrote “statistically significant value of 0.5.” That’s almost certainly a typo for 0.05. Using 0.5 makes it look like you don’t understand significance testing.

5. Report 95% confidence intervals for key metrics (accuracy, F1, precision, recall). Bootstrapping over test samples is fine if you didn’t use cross-validation.

6. Clarify how the t-test was conducted (e.g., “we compared per-class accuracies across the eight classes” or “we ran the model 5 times with different seeds and compared runs”).

7. If you cannot add cross-validation, explicitly state the reason and acknowledge it as a limitation.

8. You didn’t actually address what I asked. My concern was that your overstated clinical claims about “early diagnosis” without clinical-level validation. your answer only tells me that your datasets are expert-annotated and drops some URLs. That doesn’t solve the problem, in fact, it feels like a deflection.

The key issues (synthetic LC25000, lack of patient metadata, no outcome validation) are still unacknowledged. If you leave “early diagnosis” phrasing in the paper, I will still consider it misleading.

Providing links to Kaither and KvasirV1 is nice for transparency, but it doesn’t change the fact that these datasets are benchmarks, not clinical datasets.

your tone also feels a little dismissive. Instead of conceding that your clinical claims were too strong, you doubled down by emphasizing “verified by experts.” That’s not the same thing as real-world clinical validation.

Reviewer #2: I appreciate the authors efforts to answer the comments of reviewers. the answers were comprehensive and added value to this work

**Do you want your identity to be public for this peer review?** For information about this choice, including consent withdrawal, please see our Privacy Policy

Reviewer #1: No

Reviewer #2: **Yes: ** Hamidreza Ashayeri

---

## [Author Response · Author response to Decision Letter 2]

17 Sep 2025

Point-wise Detailed Response to Editor and Reviewers’ Comments

Title: Diagnosis of Colorectal Cancer using Residual Transformer with Mixed Attention and Explainable AI

Authors: Poonam Sharma, Bhisham Sharma, Ajit Noonia, Dhirendra Prasad Yadav, Panos Liatsis

Dear Editors and Reviewers:

We truly appreciate the time and effort you took to read our manuscript and provide valuable comments. Your comments have been extremely helpful in leading us to enhance and streamline our work. We have carefully considered your suggestions and made the necessary changes in the revised manuscript. We have made every effort to address all the points mentioned and truly hope that the revisions meet your satisfaction. The following are the changes we have made to comply with the received comments.

Reviewer Comments – 1:

Comment 1: Thank you for your responses; however, ResNeXt + ViT hybridization and mixed attention are not conceptually novel. Others have done similar hybrids. What’s the actual innovation beyond rearranging known components?

Response: The reviewer’s insightful observation is greatly appreciated. We acknowledge that previous studies have been conducted on hybridizing CNNs with transformers and attention mechanisms, such as ResNeXt. However, the conceptual innovation of our work goes beyond simply rearranging familiar components. The principal contributions of the manuscript are as follows.

1. We combine overlapping block attention (OBA) and grid attention (GA) to provide local and global attention to the spatial feature map. In traditional ViTs, multi-head self-attention (MHSA) is applied, which ignores local feature dependencies and has a quadratic computational cost. The necessity and efficacy of this design are demonstrated by the ablation study (Table 6), which shows that OBA and GA together significantly outperform MHSA.

2. Although there are ResNeXt + ViT hybrids, they usually employ standard MHSA, which is computationally costly O (M⁴D). Our suggested mixed attention block provides an accurate yet computationally efficient solution by reducing complexity to O (M²D²). The complexity analysis (Table 7) demonstrates how RNTNet achieves superior accuracy while achieving a better balance between accuracy and efficiency than more complex models like YOLOv9 and MaxViT.

3. Our model is validated on two different and clinically relevant imaging modalities, including endoscopic (KvasirV1) and histopathology images (Kaither).

4. To enhance interpretability and facilitate second opinions, we have added Grad-CAM visualizations alongside performance metrics. This integration sets our method apart from earlier work, which primarily concentrated on accuracy, and is crucial for clinical adoption.

Comment 2: State the actual slide magnification (or explicitly say if that info wasn’t available). For example, “all slides were provided at 40× magnification.”

Response: We value the reviewer’s recommendation. We have now specified the slide magnification in the updated manuscript. The Kaither NCT-CRC-HE-100K dataset comprises 224 × 224-pixel hematoxylin and eosin (H&E) stained histopathology patches extracted from whole-slide images digitized at a resolution of 0.5 µm per pixel, corresponding to approximately 20× optical magnification.

Comment 3: Trim hardware specs down to just GPU model + VRAM. The kind of detail you wrote isn’t as important for reproducibility as batch size, optimizer, etc.

Response: We are grateful for the reviewer’s recommendation. Only the GPU model and VRAM, two crucial pieces of hardware, are now documented in the experimental settings section. The important training parameters, such as batch size, optimizer, and learning rate, which are more pertinent to reproducibility, have been retained, and the rest have been removed.

Comment 4: You wrote “statistically significant value of 0.5.” That’s almost certainly a typo for 0.05. Using 0.5 makes it look like you don’t understand significance testing.

Response: We appreciate the reviewer bringing this error to our attention. In fact, the previous reference to 0.5 was a typographical error; the proper threshold for statistical significance is 0.05. To prevent any misunderstandings, we have fixed this throughout the manuscript.

Comment 5: Report 95% confidence intervals for key metrics (accuracy, F1, precision, recall). Bootstrapping over test samples is fine if you didn’t use cross-validation.

Response: Thank you for your concern. As suggested, we applied a 5-fold cross-validation scheme on the KvasirV1 and Kaither datasets and presented them in Tables 8 and 9 in the revised manuscript.

Comment 6: Clarify how the t-test was conducted (e.g., “we compared per-class accuracies across the eight classes” or “we ran the model 5 times with different seeds and compared runs”).

Response: We appreciate the reviewer’s feedback regarding the clarity of the statistical test procedure. In the revised manuscript, we have provided a clear explanation of how the paired t-test was conducted. Specifically, we employed the paired t-test for per-class performance indicators (precision and Cohen’s kappa) on the eight classes of the KvasirV1 and Kaither datasets. The model’s predicted outcomes for every class were compared to the ground truth to obtain the per-class metric values. These values were then paired and analyzed using a two-tailed paired t-test to determine if the differences between measures (such as accuracy vs. kappa) were statistically significant.

This approach was selected to ensure that the statistical test reflected both class-wise variability and aggregate performance. As made clear in the updated Results section (Statistical Analysis, Section 5.5.4), the paired t-test was conducted on per-class performance values (precision and kappa) across the eight categories of each dataset. The class-wise scores were compared and paired to see if the observed differences were statistically significant. The associated figures (Figures 9 and 10) show these results along with the mean differences and p-values.

Comment 7: If you cannot add cross-validation, explicitly state the reason and acknowledge it as a limitation.

Response: We appreciate the reviewer highlighting the value of cross-validation. We applied a 5-fold cross-validation scheme to train and test the model’s performance on the Kaither and KvasirV1 datasets. The quantitative results are depicted in Tables 8 and 9.

Table 8 displays the performance of the proposed model on the KvasirV1 dataset using 5-fold cross-validation. In Fold 1, the model showed consistent initial performance, attaining 95.18% accuracy with a precision of 94.30%, a recall of 93.89%, and an F1-score of 94.07%. Fold 2 demonstrates an accuracy of 96.40%, a recall of 94.29%, and a precision of 95.06%. Fold 3 further enhanced performance with a 96.87% accuracy. Fold 4 achieved an accuracy of 97.08% and an F1-score of 96.41%. The highest performance is observed in Fold5, with an accuracy of 98.67%, F1-score of 97.40% and Kappa of 97.14%. Overall, the average performance of all five folds is 96.80% accuracy, 95.90% precision, 95.43% recall, 95.57% F1-score, and 95.29% kappa. The results show that the proposed method performs consistently and reduces the risk of overfitting to the KvasirV1 dataset.

Table 8. Performance of the proposed model on the KvasirV1 dataset using 5-fold cross-validation

Folds Accuracy (%) Precision (%) Recall (%) F1-score (%) Kappa (%)

Fold_1 95.18 94.30 93.89 94.07 93.48

Fold_2 96.40 95.06 94.29 94.97 94.16

Fold_3 96.87 96.10 95.60 95.90 95.48

Fold_4 97.08 96.57 96.08 96.41 96.17

Fold_5 98.67 97.46 97.30 97.40 97.14

Average 96.80 95.90 95.43 95.75 95.29

The performance of the proposed model on the Kaither dataset across all five cross-validation folds is shown in Table 9. The model’s accuracy in Fold 1 was 96.06% with 95.81% precision. Fold 2 showed improvement with 97.43% accuracy, 96.28% precision, and 95.86%. Further, the Fold 3 model achieved an accuracy and precision value of 98.13% and 97.09%, respectively. Fold 4 showed steady generalization with 98.50% accuracy, 97.52% precision, 97.12% recall, 97.37% F1-score, and 97.03% Kappa. The model obtained 97.49% Kappa, 98.17% recall in Fold 5. Moreover, the model’s average precision and Kappa score are 96.98% and 96.17%, respectively.

Table 9. Performance of the proposed model on the Kaither dataset using 5-fold cross-validation

Folds Accuracy (%) Precision (%) Recall (%) F1-score (%) Kappa(%)

Fold_1 96.06 95.81 94.79 95.33 94.31

Fold_2 97.43 96.28 95.86 96.22 95.45

Fold_3 98.13 97.09 96.73 96.91 96.54

Fold_4 98.50 97.52 97.12 97.37 97.03

Fold_5 98.67 98.20 98.17 98.11 97.49

Average 97.76 96.98 96.53 96.79 96.17

Comment 8: You didn’t actually address what I asked. My concern was that your overstated clinical claims about “early diagnosis” without clinical-level validation. your answer only tells me that your datasets are expert-annotated and drops some URLs. That doesn’t solve the problem; in fact, it feels like a deflection. The key issues (synthetic LC25000, lack of patient metadata, and no outcome validation) are still unacknowledged. If you leave “early diagnosis” phrasing in the paper, I will still consider it misleading. Providing links to Kaither and KvasirV1 is nice for transparency, but it doesn’t change the fact that these datasets are benchmarks, not clinical datasets. your tone also feels a little dismissive. Instead of conceding that your clinical claims were too strong, you doubled down by emphasizing “verified by experts.” That’s not the same thing as real-world clinical validation.

Response: We sincerely thank the reviewer for this important insight. We acknowledge that our initial use of the term “early diagnosis” was very strong and could have been interpreted as implying direct clinical validation, which is not supported by the way our current study was designed. Although the Kaither and KvasirV1 datasets are annotated benchmark datasets, as the reviewer correctly points out, they do not have longitudinal outcomes, patient-level metadata, or prospective validation in real-world clinical settings. Furthermore, we did not utilize artificial datasets such as LC25000 in our research, but we completely agree that without clinical metadata and validation, the results cannot yet be applied to patient care or real-world diagnostic deployment. We modified the “early diagnosis” term to diagnosis in the revised manuscript.

In the limitations section, we have made it clear that our results are based solely on benchmark datasets and that before any claims of diagnostic value can be made, clinical-level validation on patient cohorts, including outcomes and metadata, would be necessary.

Thanks, and Regards

Authors.

---

## [Decision Letter · Decision Letter 2]

13 Oct 2025

Diagnosis of colorectal cancer using residual transformer with mixed attention and explainable AI

PONE-D-25-33937R2

Dear Dr. Panos Liatsis,

We’re pleased to inform you that your manuscript has been judged scientifically suitable for publication and will be formally accepted for publication once it meets all outstanding technical requirements.

Kind regards,

Siamak Pedrammehr, Ph.D.

Academic Editor

PLOS ONE

Additional Editor Comments (optional):

Reviewers' comments:

Reviewer's Responses to Questions

**Comments to the Author**

Reviewer #2: All comments have been addressed

Reviewer #3: All comments have been addressed

2. Is the manuscript technically sound, and do the data support the conclusions?

Reviewer #2: Yes

Reviewer #3: Yes

3. Has the statistical analysis been performed appropriately and rigorously?

Reviewer #2: Yes

Reviewer #3: Yes

4. Have the authors made all data underlying the findings in their manuscript fully available?

Reviewer #2: Yes

Reviewer #3: Yes

5. Is the manuscript presented in an intelligible fashion and written in standard English?

Reviewer #2: Yes

Reviewer #3: Yes

Reviewer #2: Use of the RNTNet with ViT and both types of overlapping block attention and grid attention are the strength of this study. I congratulate authors for this manuscript and model

Reviewer #3: Novelty Clarification: The manuscript now explicitly highlights the innovation of combining Overlapping Block Attention (OBA) and Grid Attention (GA) to achieve both local and global feature representation. The computational efficiency improvement (O(M²D²) vs. O(M⁴D) in standard MHSA) is also substantiated with comparative analysis (Table 7).

Methodological Transparency:

Preprocessing details are expanded, including image size, patch generation, and magnification (20× for Kaither dataset).

Training configuration now includes only the relevant details (GPU model + VRAM, optimizer, learning rate, batch size, and epochs), removing excessive hardware descriptions.

Validation design has been enhanced with 5-fold cross-validation on both datasets (Tables 8 and 9), along with per-fold performance reporting.

Statistical Rigor: The previously misstated threshold for statistical significance (0.5) has been corrected to 0.05. The statistical methodology is now clearly explained, including class-wise paired t-tests for performance metrics (precision and Cohen’s kappa). Confidence intervals for accuracy, precision, recall, and F1-score are reported.

Clinical Claims: The previously overstated claim of “early diagnosis” has been replaced with “diagnosis.” The limitations section now explicitly acknowledges that the results are based solely on benchmark datasets (Kaither, KvasirV1), without patient metadata or outcome validation, and that future clinical validation is required.

Language and Readability: The manuscript has been substantially revised for clarity, grammar, and professional tone. The responses to reviewers are now more precise and respectful in addressing concerns.

**Do you want your identity to be public for this peer review?** For information about this choice, including consent withdrawal, please see our Privacy Policy

Reviewer #2: **Yes: ** Hamidreza Ashayeri

Reviewer #3: **Yes: ** Saman Rajebi

---

## [Editor Report · Acceptance letter]

PONE-D-25-33937R2

PLOS ONE

Dear Dr. Liatsis,

I'm pleased to inform you that your manuscript has been deemed suitable for publication in PLOS ONE. Congratulations! Your manuscript is now being handed over to our production team.

Kind regards,

on behalf of

Dr. Siamak Pedrammehr

Academic Editor

PLOS ONE